# Therapeutic Anticancer Uses of the Active Principles of “*Rhopalurus junceus*” Venom

**DOI:** 10.3390/biomedicines8100382

**Published:** 2020-09-27

**Authors:** Mario Dioguardi, Giorgia Apollonia Caloro, Luigi Laino, Mario Alovisi, Diego Sovereto, Vito Crincoli, Riccardo Aiuto, Antonio Dioguardi, Alfredo De Lillo, Giuseppe Troiano, Lorenzo Lo Muzio

**Affiliations:** 1Department of Clinical and Experimental Medicine, University of Foggia, Via Rovelli 50, 71122 Foggia, Italy; diego_sovereto.546709@unifg.it (D.S.); antoniodioguardi@gmail.com (A.D.); alfredo.delillo@unifg.it (A.D.L.); giuseppe.troiano@unifg.it (G.T.); Lorenzo.lomuzio@unifg.it (L.L.M.); 2Department of Emergency and Organ Transplantation, Nephrology, Dialysis and Transplantation Unit, University of Bari, Via Piazza Giulio Cesare, 70124 Bari, Italy; giorgiacaloro1983@hotmail.it; 3Multidisciplinary Department of Medical-Surgical and Odontostomatological Specialties, University of Campania “Luigi Vanvitelli”, 80121 Naples, Italy; luigi.laino@unicampania.it; 4Department of Surgical Sciences, Dental School, University of Turin, 10127 Turin, Italy; mario.alovisi@unito.it; 5Department of Basic Medical Sciences, Neurosciences and Sensory Organs, Division of Complex Operating Unit of Dentistry, “Aldo Moro” University of Bari, Piazza G. Cesare 11, 70124 Bari, Italy; vito.crincoli@uniba.it; 6Department of Biomedical, Surgical, and Dental Science, University of Milan, 20122 Milan, Italy; riccardo.aiuto@unimi.it

**Keywords:** *Rhopalurus junceus*, blue scorpion, venom, vidatox, cancer

## Abstract

The *Rhopalurus junceus* is a scorpion belonging to the Buthidae family that finds its habitat in Cuba. This scorpion is known by the common name of “Blue Scorpion”. The venom is used on the island of Cuba as an alternative cure for cancer and, more recently, in the research of active components for biomedicine. Recently, the venom has been tested in several studies to investigate its effects on cancer cell lines, and the initial results of in vitro studies demonstrated how this poison can be effective on certain carcinoma cell lines (Hela, SiHa, Hep-2, NCI-H292, A549, MDA-MB-231, MDA-MB-468, and HT-29). The aim of this review is, therefore, to describe the effects of the venom on carcinoma lines and to investigate all anti-cancer properties studied in the literature. The research was conducted using four databases, Pub Med, Scopus, EBSCO, and Web of Science, through the use of keywords, by two independent reviewers following the PRISMA protocol, identifying 57 records. The results led to a total of 13 articles that met the eligibility criteria. The data extracted for the purpose of meta-analysis included the IC_50_ of the venom on carcinoma cell lines. The results of the meta-analysis provided a pooled mean of the IC_50_ of 0.645 mg/mL (95% CI: 0.557, 0.733), with a standard error (SE) = 0.045, *p* < 0.001. The analysis of the subgroups, differentiated by the type of cell line used, provided insight regarding how the scorpion venom was effective on the cell lines of lung origin (NCI-H292, A549, and MRC-5) with a pooled mean of IC50 0.460 mg/mL (95% CI: 0.290, 0.631) SE (0.087) *p* < 0.001. The results described in the literature for in vitro studies are encouraging, and further investigations should be carried out and deepened.

## 1. Introduction

*Rhopalurus junceus* is a scorpion belonging to the Buthidae family that finds its habitat in Cuba. The venom is used in the Cuban islands as an alternative cure for cancer and, more recently, in the search for active components in biomedicine. This scorpion is known by the common name of “Blue Scorpion”, and Escozul is a diluted solution of its venom.

The first article present in the international scientific literature in the medical field, which investigated the anti-tumor properties, is a communication by Garcia-Gomez et al. 2011 [1]. Garcia-Gomez concluded that the venom was not dangerous for mammals (tested on mice) but was lethal to crickets (strain Acheta domestica), and it contained phospholipases and hyaluronidases with antimicrobial activities and had an action directed toward the currents of Na^+^- and K^+^-ion channels in neuroblastoma cells. Other studies from the same research group investigated the toxic potential of the venom aimed at cancerous cell lines, noting an efficacy directed toward epithelial cancerous cell lines [2]. The cancer cell lines primarily studied were hematopoietic and epithelial lines (pulmonary, mammary, hepatic, and cervix). Recent studies of 2019 and 2020 investigated the pharmacokinetics, pharmacodynamics, and teratogenic effects of the venom [3,4].

There are currently no literature reviews on the systemic effects of this poison. In light of these discoveries and in view of the ever increasing need to identify chemotherapies in the neoplastic field, in this review we aimed to summarize the results of the studies of the last decade, inherent in the anticancer effects, of the venom of *Rhopalurus junceus* (blue scorpion).

## 2. Materials and Methods

The study was conducted with reference to the guidelines described by the PRISMA (Preferred Reporting Items for Systematic Review and Meta-Analysis) [5]. The materials and methods used in this review were also used in previous reviews of the same author’s literature [6,7]. The questions the authors ask are: “What are the effects of scorpion venom on cancer cell lines?” and, more generally, “What are the therapeutic potentials of the poison?” Considering these questions, the search for bibliographical sources was conducted.

After an initial selection phase, in which the records were identified in the databases, potentially suitable articles were qualitatively assessed with the aim of identifying any useful information that described the effects and potential of the blue scorpion venom.

Scientific studies that investigated the therapeutic effects of scorpion venom extracts were considered potentially eligible—in particular, all studies that investigated the effects of the poison on normal and carcinomatous cell lines. Secondly, we also researched those articles that could have any information on the composition of the poison, on its bioavailability, and on potential in vivo effects or in animal models, conducted in recent years (10 years), published in English, and indexed in international databases of medicine.

We decided to choose articles published in the last 10 years because researchers began to investigate the possible use of the poison of *Rhopalurus junceus*, starting in 2011. The potentially eligible articles were finally subjected to a full-text analysis in order to verify their use for a qualitative and quantitative analysis. The inclusion and exclusion criteria applied in the full-text analysis are as follows:We included all those studies that reported data on the effects on cancer cell lines; the composition of the extracted venom; the bioavailability; the teratogenic, toxic, and therapeutic effects.The studies were excluded if there was no reported relevant data in the biomedical field, if they were not written in English, or if they were not published in international medical journals.

### 2.1. Research and Screening Methodology

The articles were identified using the electronic databases, Pub Med, Scopus, Web of Science, and EBSCO, and the bibliographical references of the articles were consulted to potentially identify further studies to include.

The research of the studies was conducted between 15 July 2020 and 30 July 2020.

The following search terms were used in Pub Med, Scopus, EBSCO, and Web of Science searches: Rhopalurus junceus (Pub Med 12, Scopus 14, EBSCO 11, and Web of Science 9), Vidatox (Pub Med 2, Scopus 2, and EBSCO 2) (Table 1). To complement this research, we conducted a manual evaluation of the articles included in the references of the identified full-text publications and no citations were deemed relevant.

The reviewers responsible for the research and screening of the records were two reviewers (M.D. and D.S.). The search was conducted independently, and the keywords and databases to be used were preliminarily agreed upon. A third reviewer, to the end of the search for potentially eligible studies, decided on any doubtful situations (G.T.).

The screening was performed by reading the title and the abstract to eliminate any records not related to the subject of the review. In the case of the doubt of the individual reviewer, the full text of the article was read. The articles obtained were subjected to full-text analysis by the two reviewers (13 articles), from which those eligible for the qualitative analysis and inclusion in the meta-analysis for the result were identified. The results sought by the two reviewers were the following: Primary outcome—the effects of *Rhopalurus junceus* venom on cancer line cells (specifically IC_50_, venom concentration that causes 50% reduction in cells) was researched.

### 2.2. Statistical Analysis Protocol

For the quantitative analysis, following the extraction of the data of the included studies, a statistical analysis of the data was conducted, which included a meta-analysis of the effects of the venom on the cell lines on which it was effective (primary outcome). A subgroup analysis was also conducted (the groups differed by the type of cell line tested). The subgroups analyzed were as follows: cervix, lung mammary, gland/breast, and colorectal adenocarcinoma. The mean and 95% confidence intervals of the IC_50_ values were calculated for each individual study in order to estimate the pooled mean of the venom’s effect on the cancer cell line was calculated for the meta-analysis.

The protocol with which the meta-analysis was performed was based on the recommendations of the Cochrane Manual for Systematic Reviews of Interventions. We decided to use Open Meta-Analyst version 10 (Tufts University, Medford, MA, USA) as the meta-analysis software. The presence of heterogeneity was evaluated by calculating the Higgins index (*I*^2^), using the Open Meta-Analyst software. If the measurement turned out to be greater than 50%, the heterogeneity rate was considered high. The results of the meta-analysis were represented by a forest plot.

## 3. Results

From the database searches (Scopus, Pub Med, Web of Science, and EBSCO), we identified 57 records. With the use of the EndNote software, the duplicates were removed, resulting in 19 records. With the inclusion and exclusion criteria applied, we maintained 13 articles, of which only 7 dealt with the effects of scorpion venom on cancer cell lines and only 4 studies reported the data used in the meta-analysis. All selection and screening procedures are described in the flow chart, shown in Figure 1.

The studies included for qualitative analysis were the following: Garcia-Gomez et al. 2011 [1], Rodriguez-Ravelo et al. 2013 [8], Diaz-Garcia et al. 2013 [2], Rodriguez-Ravelo et al. 2015 [9], Diaz-Garcia et al. 2015 [10], Giovannini, C. et al. 2017 [11], Diaz-Garcia et al. 2017 [12], Diaz-Garcia et al. 2019 [3], Lagarto et al. 2020 [4], Diaz-Garcia et al. 2019 [13], Di Lorenzo et al. 2012 [14], Yglesias-Rivera et al. 2019 [15], and Lozano-Trujillo et al. 2020 [16].

We included seven studies in the quantitative analysis: Diaz-Garcia et al. 2013 [2], Diaz-Garcia et al. 2015 [10], Giovannini et al. 2017 [11], Diaz-Garcia et al. 2017 [12], Diaz-Garcia et al. 2019 [13], Yglesias-Rivera et al. 2019 [15], and Lozano-Trujillo et al. 2020 [16]. Four of these were the studies from which the data for the meta-analysis were extracted (Table 2).

The extracted data included the type of cell lines tested, the IC_50_ of the venom (mean and standard deviation), the number of times the test was repeated, the method of extraction of the poison, any other results obtained, and information on the article (date, first author). The results are summarized in Table 2 and Table 3.

From Table 2, it is clear that the cell lines on which the venom was effective are mainly those of epithelial origin (lung, cervix, and mammary glandular). The hematopoietic lines were not affected by the effect of the venom of the scorpion as well as the non-cancer cell lines.

The method of collecting the poison was identical in all four studies, performed with electrical stimulation, filtered and centrifuged, and separated from the supernatant before finally being stored at a temperature of −20 degrees centigrade. In Diaz-Garcia et al. 2015 [10], the venom was also divided into five parts.

The studies presented in Table 3 were excluded from the meta-analysis. Giovannini et al. 2017 reported the effects of Vidatox 30, which is the hematopoietic derivative of venom [11]. Yglesias-Rivera et al. 2019 reported data regarding a synergistic effect between antineoplastic drugs and scorpion venom. The data could, therefore, not be used for the purpose of meta-analysis [15]. The study by Lozano-Trujillo et al. 2020 reported the data of the IC_50_ without the standard deviation, which was, therefore, not usable for the purpose of meta-analysis [16].

### 3.1. Risk of Bias

The risk of bias within the studies was assessed by the first reviewer (M.D.) and for the four studies they were relatively low; the risk of bias between studies was also low as the four studies were performed by the same research group with a similar study design. Despite everything, there were differences that make the results of the studies heterogeneous with each other; in fact, the heterogeneity between the results stands at *I*^2^, equal to 99.02%. The reasons for this heterogeneity are found in the different cell lines tested with the poison and, in some studies, the fraction of the poison used.

To minimize the heterogeneity in the results, we decided to perform a subgroup analysis (cervix, lung mammary, gland/breast, and colorectal adenocarcinoma), the subgroup analysis demonstrated a very slight reduction in heterogeneity with values for the subgroup: cervix *I*^2^ equal to 99.35%, lung *I*^2^ equal to 99.3%, and breast *I*^2^ equal to 93.54%. In addition, the high heterogeneity was confirmed by the graphic analysis of the confidence intervals of the forest plot, which demonstrates the low overlap in all the subgroups.

### 3.2. Meta-Analysis

Statistical data analysis was performed using Open Meta-Analyst version 10 software (Tufts University, Medford, MA, USA).

The meta-analysis showed an absence of heterogeneity with *I*^2^ equal to 99.702% with Q = 5030.957 df 15. *p* <0.001, and a random effect model was applied. The results shown in Figure 2 show that a pooled mean of IC_50_ was obtained at 0.645 mg/mL (95% CI: 0.557, 0.733), standard error (SE) = 0.045, *p* < 0.001.

The analysis of the subgroups reports the following results (Figure 3):Subgroup Cervix: pooled mean of IC_50_ 0.917 mg/mL (95% CI: 0.799, 1.035) SE (0.060) *p* < 0.001 *I*^2^ 99.35%;Subgroup Lung: pooled mean of IC_50_ 0.460 mg/mL (95% CI: 0.290, 0.631) SE (0.087) *p* < 0.001 *I*^2^ 99.3%;Subgroup mammary gland/breast: pooled mean of IC_50_ 0.699 mg/mL (95% CI: 0.640, 0.758) SE 0.030 *p* < 0.001 *I*^2^ 93.54%;Subgroup colorectal, adenocarcinoma: IC_50_ mg/mL 0.890 (95% CI: 0.872, 0.908) SD (Standard Deviation) 0.020.

## 4. Discussion

Diaz-Garcia et al. 2013 [2] led the first study on multiple epithelial tumor cell lines, (Hela, SiHa, Hep-2, NCI-H292, A549, MDA-MB-231, MDA-MB-468, and HT-29), hematopoietic (U937, K562, and Raji), and non-tumor cell lines (MRC-5 and MDCK), and the results indicated the sensitivity of the scorpion venom toward epithelial tumor cell lines (more sensitive toward A549 and less sensitive toward Hela). We found that epithelial cancer cells, the lung (A549 and NCI-H292) and breast cell lines (MDA-MB-213 and MDA-MB-468) were slightly more sensitive [2]. Less than 50% cell viability was found for scorpion venom between 0.1–0.75 mg/mL of cells. There was 50% viability reduction only at concentrations of poison higher than 0.75 mg/mL for the remaining cell lines [2].

The effects on cell lines of pulmonary origin were confirmed in a 2015 study in which the venom, fractionated into its components (five fractions according to their weight), showed different efficacies. The FI fraction induced a decrease in the cell viability in MRC-5 more than in A549. The FII and FIII did not show differences between cancer and normal cells. The FIV and FV differences were significantly lower [10].

The effects on the breast cell line also came from a 2017 study by Diaz-Garcia et al. [12], who reported that venom inhibited the growth of triple negative breast cancer cells, MDA-MB-231, and regulated the expression of apoptosis-related genes, which induced apoptosis via the mitochondrial–apoptotic pathway. Similar results were obtained on a cell line (F3II mammary adenocarcinoma cell) in a 2019 study, concluding that *Rhopalurus junceus* scorpion venom was able to induce apoptotic cell death against murine breast cancer cells in vitro and effectively inhibited mammary tumor progression in mice [12].

Another consideration is to be made on the ineffectiveness described by Giovannini et al. 2017 [11], for the homeopathic drug Vidatox 30-CH, which is an alcoholic solution at 33% resulting from five low molecular weight peptides (Escozul). The drug was used on a hepatic carcinoma cell line (HepG2-Snu449 and BRL-3A cells) resulting in ineffectiveness in inhibiting proliferation and an absence of a synergistic effect with the drug sorafenib in rats with hepatocellular carcinoma (HCC). Treatment with Vidatox for 30 days increased the tumor growth more in rats with HCC than in the control rats [11].

Synergistic effects were found instead between blue scorpion venom (0.5 mg/mL) in a HeLa cancer cell line when combined with 5-fluorouracil (0.5–500 µM), cisplatin (3.13–25 µM), or doxorubicin (3.13–25 µM) in an experiment conducted by Yglesias-Rivera et al. 2019 [15].

A recent study published in 2020 also reported the effects on glioblastoma cell lines, indicating that a venom concentration of 50 μg/mL was capable of decreasing the viability of T98G cells at levels significantly lower than the previous studies conducted on different cell lines [16].

Studies on the molecular components of the venom conducted by Rodriguez-Ravelo et al. investigated the different composition of scorpion venom in different geographical areas of Cuba from five sites (Moa, La Poa, Limonar, El Chote, and Farallones), identifying no substantial difference in the composition [8]. A subsequent study of the same group found no substantial difference between female and male scorpions in composition and identified (172–193) components [9].

This poison causes few adverse symptoms on humans. Diaz-Garcia et al. 2015 found caseinolytic, gelactinase, and hemolytic activity with a toxic effect on the gastrocnemius muscles with the release of CK and blood LDHA only at high concentrations (at concentrations of 12, 5, and 25 mg/kg). Again, in this study they identified that low molecular weight fractions (<4 kDa) induced significant cytotoxicity in A549 cells (lung cancer) while high molecular weight proteins (45–60 kDa) were responsible for the hyaluronidase activity and the toxic effect against MRC-5 [10].

Through a pharmacokinetic study on mice carrying tumors, we identified that the venom was rapidly eliminated from the body and had a greater localization for tumor tissues; moreover, the intravenous route compared to the oral one appeared to increase the bioavailability of the drug [13]. Lagarto et al. 2020 [4] reported that oral administration in mice of up to 100 mg/kg/day during the period of organogenesis did not induce maternal and embryo–fetal toxicity, which had a low impact on reproductive physiology and did not affect the health of the animals.

Clinical trials on the use of *Rhopalurus junceus* venom as an antitumor agent have not been published and are not currently present in international medical scientific journals. Di Lorenzo et al. 2012 reported a case report on a patient with gastric cancer who felt the benefits of the venom, with an improvement in the quality of life and a reduction in pain as well as an unexpected reduction in prostate-specific antigen (PSA) [14].

### 4.1. Mechanisms of Action of Venom on Cell Lines

The venom’s mechanism of action may be associated with peptides that recognize sodium and potassium channels, and the venom’s selective cytotoxic effect could also be associated with its effect on inducing apoptosis via the mitochondrial pathway [12]. There are indications that scorpion venom may have real negative effects on tumor neo-angiogenesis [13].

On the other hand, research demonstrated that cell lines, such as Hela and MDA-MB-231, were prone to cell death when treated with *Rhopalurus junceus* venom by mechanisms associated with the increased expression of apoptotic genes, such as *Bax*, *Noxa,* and *Puma*; the mechanism of action involved the upregulation of the *bax*, p53, and caspase 3 (the gene expression of bax and caspase 3 increased significantly after 24 h, and that of p53, after 48 h, on the cell lines of F3II cancer cells) genes; the downregulation of the *bcl*-2 genes likely as a result of the upregulation of p53. This included the activation of caspases, chromatin condensation, and the formation of apoptotic nuclei, in Diaz 2013 [2].

The increase in p53 led to an upregulation of bax and caspase 3 and to a downregulation of bcl-2. The increase in the bax/bcl-2 ratio activated the mitochondrial pathway of apoptosis with the release of cytochrome c (Cyt c) and the activation of caspase cascades (3, 6, 7), which led to the proteolytic breakdown of important cytoplasmic and nuclear substrates [17].

The role of ion channels in the tumor adhesion, proliferation, and invasion processes is known [18]. The interaction of ion channels is fundamental in the processes of changing the volume and cell shape, as well as the glioma, through variations in the flow of Cl^−^ and K^+^ ions, while variations in the concentration of Ca^+^ varied the shape and volume, allowing the neoplastic cell to invade the surrounding tissues [19].

There was a confirmation that the venom of the blue scorpion led to a reduction in the expression of ki-67 and CD31, whose overexpression was associated with tumor angiogenesis.

These processes occur through the interaction of ion channels with changes in the cell morphology and volume. In particular, this was demonstrated to be the mechanism of the growth and invasion of glioma cells, mainly through the electrochemical efflux of Cl^−^ and K^+^, such as KCl and water, which is mediated by an increase in intracellular Ca^2+^ ions, which leads to cell shrinkage and allows the cell to invade brain tissue [19].

Therefore, the mechanism of action with which the poison of *R. junceus* could act against neoplasms could be through three mechanisms: a mitochondrial apoptotic action through the intrinsic pathway with an increase in the expression of p53 and an increase in the bax-BCL-2 ratio; the blocking action of the Na K and Cl canal, which would act on the concentration of intracellular calcium with a reduction in the capacity of variation of the shape and volume and, therefore, of tissue invasion; an influence on the CD 31- and Ki-67-mediated neoangiogenesis processes [20].

The histological examination of tumor mass in studies on animal models also highlighted how the scorpion venom induced an extensive necrosis in the tumor compared to the tissue from the evidence of the untreated control group [4].

There was also an antimicrobial effect carried out by the presence of phospholipase and hyaluronidase in the venom with antibacterial action [1].

### 4.2. Quantitative Analysis Discussion

The analysis of the articles included in the quantitative assessment confirmed that the blue scorpion venom was more effective on cell lines of epithelial glandular origin with the exception of the latest study reported by Lozano-Trujillo on glioblastoma lines.

The meta-analysis of the results provided an average pooled IC_50_ (venom concentration that causes 50% reduction in cells) of 0.645 mg/mL. The analysis of the subgroups showed that the venom had a greater efficacy in IC_50_ against cell lines of lung origin (NCI-H292, A549, and MRC-5) with a pooled mean of IC_50_ 0.460 mg/mL compared to mammary glandular cell lines (MDA-MB-231, F3II, and MDA-MB-468) with a pooled mean of IC_50_ 0.699 mg/mL. All subsequent studies tended to confirm the first cell line study conducted by Diaz-Garcia et al. 2013 [2].

The cell lines that were not affected by the venom were those of hematopoietic origin: U937 (histiocytic lymphoma), K562 (chronic myelogenous leukemia), Raji (Burkitt’s lymphoma), and certain non-neoplastic lines: MDCK (normal canine kidney), Vero (normal African green monkey kidney), and BALB/3T3 (normal fibroblast).

### 4.3. Future Research Lines

The next steps to be taken in light of the results obtained must be directed toward a greater confirmation of the efficacy on cell lines in which the preliminary data reported an efficacy at very low concentrations (for example, glioblastoma cell lines with IC_50_ 75 μg/mL) [16], and then to move on to studies on models animals, to verify their efficacy not only on single cells but also on growing tumor masses. At the same time, the identification within the extracted poison of the bioactive molecular components responsible for the supposed antitumor effects is of fundamental importance.

## 5. Conclusions

The evidence from the literature demonstrated how the venom of the scorpion *Rhopalurus junceus* is cytotoxic toward cancer cell lines of epithelial origin at a concentration given by the meta-analysis equal to 0.645 mg/mL. There were no data in the scientific literature showing that the homeopathic alcoholic dilution version (Escozul) was effective for both in vitro and animal models. Blue scorpion venom could represent an interesting source of chemotherapy drugs and its properties should be further investigated.

## Figures and Tables

**Figure 1 biomedicines-08-00382-f001:**
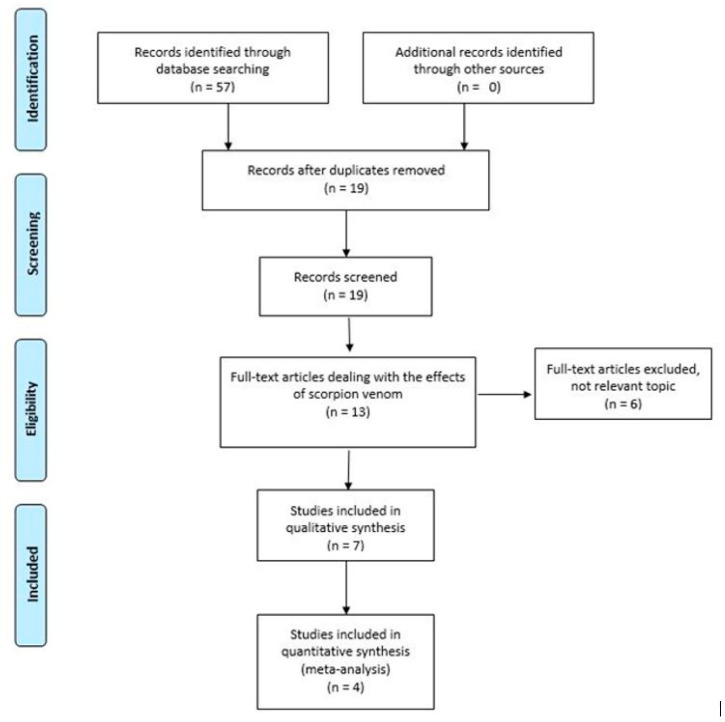
Flowchart of the different phases of the systematic review. Identification: 57; Screening: 19; Eligibility 13; Included 7; Meta-analysis 4.

**Figure 2 biomedicines-08-00382-f002:**
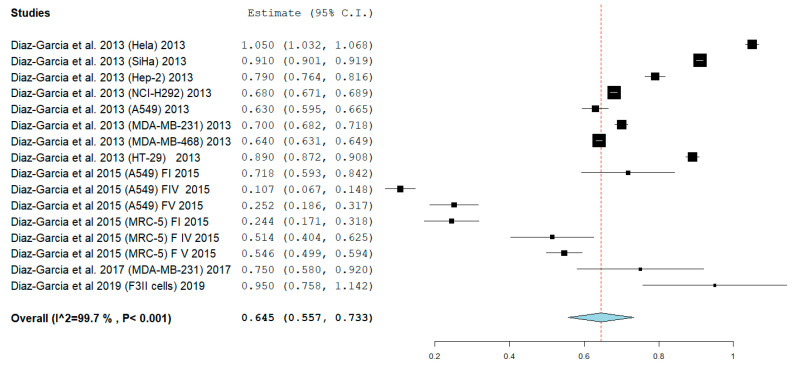
The random effect meta-analysis (Q 5030.957, df 15, I2 99.702%, *p* < 0.001) IC_50_ 0.645 mg/mL (95% CI: 0.557, 0.733), standard error (SE) = 0.045, *p* < 0.001. Legend: Q = Q statistic (measure of weighted squared deviations); df = degrees of freedom; *I*^2^ (I^2) = Higgins heterogeneity index, *I*^2^ < 50%, heterogeneity irrelevant; *I*^2^ > 75%, significant heterogeneity; C.I. = confidence intervals; P = *p* value. The graph for each study shows the cell lines investigated, the first author, and the date of publication, as well as the measurement of the IC_50_ expressed in mg/mL with the confidence intervals reported. The final value is expressed in bold with the relative confidence intervals. The red line shows the position of the average value and the rhombus in light blue shows the measure of the average effect.

**Figure 3 biomedicines-08-00382-f003:**
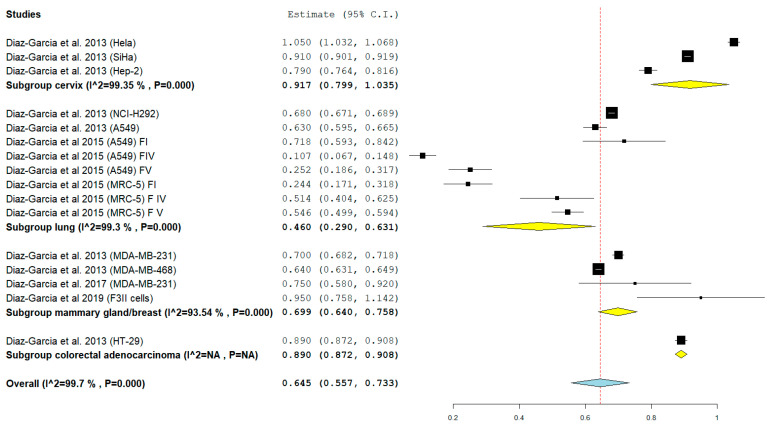
Forest plot of the 4 Subgroups. Subgroup Cervix: IC_50_ 0.917 mg/mL (Hela, SiHa, Hep-2); Subgroup Lung: IC_50_ 0.460 mg/mL (NCI-H292, A549, MRC-5); Subgroup mammary gland/breast: IC_50_ 0.699 mg/mL (MDA-MB-231, F3II, MDA-MB-468); Subgroup colorectal, adenocarcinoma: IC_50_ mg/mL 0.890 (HT-29). The results of the meta-analysis for each subgroup are highlighted in bold. Yellow rhombuses in the forest plot indicate the average effect for each subgroup investigated, the red line shows the position of the average value and the rhombus in light blue shows the measure of the average effect. NA stands for not applicable.

**Table 1 biomedicines-08-00382-t001:** Complete overview of the search methodology. Records identified by databases: 57. The overlaps were removed using the EndNote 8 software, and there were four articles selected for quantitative analysis.

Database-Provider	Keywords	Search Details	Number of Records	Number of Records after Restriction by Year of Publication (Last 10 Years)	Articles after Removing Overlapping Articles	Number of Remaining Articles Related to the *Rhopalurus junceus*	Number of Articles Remaining after Applying the Inclusion and Exclusion Criteria	Number of Articles Dealing with the Effects of Scorpion Venom on Cell Lines	Number of Articles Included for the Quantitative Analysis
Pub-med	Rhopalurus junceus	“Rhopalurus” (All Fields) AND “junceus” (All Fields)	12	12					
Pub-med	Vidatox	“Vidatox” (All Fields)	2	2					
Scopus	Rhopalurus junceus	TITLE-ABS-KEY (rhopalurus AND junceus)	14	14					
Scopus	Vidatox	TITLE-ABS-KEY (vidatox)	6	6					
EBSCO	Rhopalurus junceus	Boolean/Phrase: Rhopalurus junceus	12	12					
EBSCO	Vidatox		2	2					
Web of Science	Rhopalurus junceus		9	9					
Other Bibliographic sources (literature reviews)			0	0					
Total records			57	57	19	19	13	7	4

**Table 2 biomedicines-08-00382-t002:** The data extracted for the four articles included in the meta-analysis.

First Autor, Data, Reference	Cell Lines	Tissue of Cell Origin	IC_50_ (Mean, SD)	Number of Repeated Tests	No Effect	Venom	Result
Diaz-Garcia et al. 2013 [2]	Hela	Human cervix epitheloid carcinoma	IC_50_ 1.05 ± 0.02 (mg/mL)	5	MRC-5 (normal human lung fibroblast), MDCK (normal canine kidney), Vero (normal African green monkey kidney), U937 (histiocytic lymphoma), K562 (chronic myelogenous leukemia), Raji (Burkitt’s lymphoma)	Venom from scorpions kept alive in the laboratory was extracted by electrical stimulation. Venom was dissolved in distilled water and centrifuged. The supernatant was filtered and stored at −20 °C.	The result evidenced a significant reduction in cell viability against epithelial cancer cell lines (carcinomas and adenocarcinomas) when compared to respective control cells (MRC-5, MDCK, Vero). Venom did not affect the viability in normal hematopoietic cell lines (U937, K562, Raji).
	SiHa	cervix squamous cell carcinoma	IC_50_ 0.91 ± 0.01 (mg/mL)				
	Hep-2	cervix carcinoma	IC_50_ 0.79 ± 0.03 (mg/mL)				
	NCI-H292	mucoepidermoid pulmonary carcinoma	IC_50_ 0.68 ± 0.01 (mg/mL)				
	A549	human lung carcinoma	IC_50_ 0.63 ± 0.04 (mg/mL)				
	MDA-MB-231	mammary gland/breast	IC_50_ 0.7 ± 0.02 (mg/mL)				
	MDA-MB-468	mammary gland/breast	IC_50_ 0.64 ± 0.01 (mg/mL)				
	HT-29	colorectal adenocarcinoma	IC_50_ 0.89 ± 0.02 (mg/mL)				
Diaz-Garcia et al. 2015 [10]	A549	human lung carcinoma	IC_50_ 717.65 ± 110 (µg/mL) FI	3		The crude venom was separated into five selected fractions (I–V).	The FI fraction induced a decrease in the cell viability in MRC-5 more than in A549. The FII and FIII did not show differences between cancer and normal cells. The FIV and FV differences were significantly lower.
			IC_50_ 107.4 ± 36 (µg/mL) FIV				
			IC_50_ 251.6 ± 58 (µg/mL) FV				
	MRC-5	normal human lung fibroblast	IC_50_ 244.5 ± 65 (µg/mL) FI				
			IC_50_ 514.5 ± 98 (µg/mL) FIV				
			IC_50_ 546.5 ± 42 (µg/mL) FV				
Diaz-Garcia et al. 2017 [12]	MDA-MB-231	mammary gland/breast	IC_50_ 0.75 ± 0.15 (mg/mL)	3	Vero (normal African green monkey kidney	Venom from scorpions kept alive in the laboratory was extracted by electrical stimulation. Venom was dissolved in distilled water and centrifuged. The supernatant was filtered and stored at −20 °C.	Inhibits the growth of breast cancer cells MDA-MB-231. Regulated the expression of apoptosis-related genes, inducing apoptosis through the mitochondrial-apoptotic pathway. Vero cells were affected minimally in morphology and viability at the highest concentration used in the study.
Diaz-Garcia et al. 2019 [13]	F3II cells	mouse mammary sarcomatoid carcinoma	IC_50_ 0.95 ± 0.17 (mg/mL)	3	BALB/3T3 (normal Fibroblast)	Venom from scorpions kept alive in the laboratory was extracted by electrical stimulation. Venom was dissolved in distilled water and centrifuged. The supernatant was filtered and stored at −20 °C.	*Rhopalurus junceus* scorpion venom was able to induce apoptotic cell death against murine breast cancer cells in vitro and effectively inhibited the mammary tumor progression.

**Table 3 biomedicines-08-00382-t003:** The data extracted from articles not included in the meta-analysis (NE = no effect).

First Author, Data, Reference	Cell Lines	Tissue of Cell Origin	Effect	Venom	Results
Giovannini et al. 2017 [11]	HepG2	Human hepatocyte carcinoma	NE	Vidatox 30-CH that is an alcoholic solution at 33% resulting from five low molecular weight peptides extracted from the blue scorpion venom.	Rats with HCC, treatment with Vidatox for 30 days achieved advanced growth cancer more than in control rats.
	Snu449	Hepatocellular carcinoma	NE		
	BRL-3A	Rat liver cell	NE		
Yglesias-Rivera et al. 2019 [15]	HeLa	Human cervix epitheloid carcinoma	5-fluoruracil (0.5–500 µM) or Cisplatin (3.13–25 µM) or doxorubicin (3.13–25 µM) and scorpion venom (0.5 mg/mL)	Venom from scorpions kept alive in the laboratory was extracted by electrical stimulation. Venom was dissolved in distilled water and centrifuged. The supernatant was filtered and stored at −20 °C.	Scorpion venom exerted synergic effects in a HeLa cancer cell line when combined with 5-fluorouracil, cisplatin, and doxorubicin.
	vero	Vero (normal African green monkey kidney	NE		
Lozano-Trujillo et al. 2020 [16]	T98G	Glioblastoma Cell Line	IC_50_ 75 μg/mL	The venom was obtained from wild species by electrical stimulation and centrifuge. The supernatant was filtered and stored at −80 °C.	Results demonstrated that the venom concentration of 50 μg/mL was capable of diminishing the viability of T98G cells.

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
