# Peer review of "Therapeutic Anticancer Uses of the Active Principles of “Rhopalurus junceus” Venom"

_biomedicines, 2020, doi:10.3390/biomedicines8100382_

Round 1
Reviewer 1 Report
This review on therapeutic anticancer uses of the active principles of the “Rhopalurus junceus” Venom although interesting but has very limited information in literature.
I feel this review will encourage researcher in the filed of cancer especially the ones working on use of venom as anti-cancer agent to further investigate Rhopalurus junceus and study its effects both in-vivo and in-vitro.
Comments to authors;
- clearly mention a pathway via which this venom is or may be working.
- Provide a plan for future research to guide researcher to carry out work in this area.
Author Response
Reviewer 1
This review on therapeutic anticancer uses of the active principles of the “Rhopalurus junceus” Venom although interesting but has very limited information in literature.
I feel this review will encourage researcher in the filed of cancer especially the ones working on use of venom as anti-cancer agent to further investigate Rhopalurus junceus and study its effects both in-vivo and in-vitro.
Comments to authors;
- clearly mention a pathway via which this venom is or may be working.
- Provide a plan for future research to guide researcher to carry out work in this area.
Answer
thanks for the suggestions and comments.
the following paragraphs have been added to the manuscript as suggested
- Mechanisms of Action of Venom on cell lines
The venom's mechanism of action may be associated with peptides that recognize sodium and potassium channels and venom's selective cytotoxic effect would also be associated with its effect of inducing apoptosis via the mitochondrial pathway [12]. Furthermore, there are indications that scorpion venom may have real negative effects on tumor neo-angiogenesis [13].
On the other hand, it has been shown that cell lines such as Hela and MDA-MB-231 are prone to cell death when treated with Rhopalurus junceus venom by mechanisms associated with increased expression of apoptotic genes such as Bax, Noxa and Puma, the mechanism action involved the upregulation of the bax a p53 and caspase 3 (gene expression of bax and caspase 3 would increase significantly after 24 hours while that of p53 after 48 hours, on cell lines F3II cancer cells) genes and the downregulation of the bcl-2 genes likely as a result of the upregulation of p53 and included the activation of caspases, chromatin condensation and the formation of apoptotic nuclei Diaz 2013 [2].
The increase in p53 would lead to an up regulation of bax and capsase 3 and to a down regulation of bcl-2. The increase of the bax / bcl-2 ratio activates the mitochondrial pathway of apoptosis with the release of cytochrome c (Cyt c) and the activation of caspase cascades (3 6 7) which lead to the proteolytic breakdown of important cytoplasmic substrates and nuclear.
The role of ion channels in tumor adhesion, proliferation and invasion processes is known. The interaction of ion channels is fundamental in the processes of changing the volume and cell shape, the glioma in fact through variations in the flow of Cl-and K+ ions with variations in the concentration of Ca+ varies its shape and volume, allowing the neoplastic cell to invade surrounding tissues.
Furthermore, there is confirmation that the venom of the blue scorpion leads to a reduction of the expression of ki-67 and CD31 whose overexpression are associated with tumor angiogenesis.
These processes occur through the interaction of ion channels with changes in cell morphology and volume. In particular, this has been shown to be the mechanism of the growth and invasion of glioma cells mainly through the electrochemical efflux of Cl- and K + such as KCl and water, which is mediated by an increase in intracellular Ca + 2 ions , which leads to cell shrinkage which allows the cell to invade brain tissue.
Therefore, the mechanism of action with which the poison of R. junceus could act against neoplasms could be through 3 mechanisms: a mitochondrial apoptotic action through the intrinsic pathway with an increase in the expression of p53 and an increase in the bax-BCL-2 ratio; The blocking action of the Na K and Cl channels which would act on the concentration of intracellular calcium with a reduction in the capacity of variation of the shape and volume and therefore of tissue invasion; and an influence on CD 31 and Ki-67 mediated neoangiogenesis processes.
From the histological examination of tumor mass in study on animal models it is also highlighted how the scorpion venom induced an extensive necrosis in the tumor compared to the tissue from the evidence of the untreated control group.
There is also an antimicrobial effect carried out by the presence in the venom of phospholipase and hyaluronidase with antibacterial action.
- Future Research Lines
The next steps to be taken in light of the results obtained must be directed towards a greater confirmation of the efficacy on lines in which the preliminary data report an efficacy at very low concentrations (for example Glioblastoma Cell line with IC50 75 μg /mL) [16], to move on to studies on models animals, to verify their efficacy not only on single cells but also on growing tumor masses, at the same time it is of fundamental importance the identification, in the extracted poison, of the bioactive molecular components responsible for the supposed antitumor effects
Reviewer 2 Report
Dioguardi et al. have reported the manuscript which title is “Therapeutic anticancer uses of the active principles of the “Rhopalurus junceus” Venom”. The manuscript has shown the anit-cancer effect of Rhopalurus junceus Venom. However, there are only one study group and their 4 publication in the manuscript. The research population should be larger to convince readers of the results of this research. In addition, the authors should reorganize the paragraph of discussion and methods. And, the authors should show the figure legends more detail.
Author Response
Reviewer 2
Dioguardi et al. have reported the manuscript which title is “Therapeutic anticancer uses of the active principles of the “Rhopalurus junceus” Venom”. The manuscript has shown the anit-cancer effect of Rhopalurus junceus Venom. However, there are only one study group and their 4 publication in the manuscript. The research population should be larger to convince readers of the results of this research. In addition, the authors should reorganize the paragraph of discussion and methods. And, the authors should show the figure legends more detail.
Answer
thanks for the suggestions and comments
- The studies included in the systematic review are 7 and not 4 studies, the meta-analysis data come from 4 studies of the same research group(Diaz-Garcia et al) , but the conclusions of the systematic review are drawn from 7 studies included in 4 different study groups( Giovannini et al, Yglesias-Rivera et al, Lozano-Trujillo et al, Diaz-Garcia et al.)
- The Discussion and Methods sections have been changed and reorganized
- the legends of the figures have been reported in more detail
Figure 1. Flowchart of the different phases of the systematic review. Identification: 57; Screening: 19; Eligibility 13; Included 7; Meta-analysis 4.
Figure 2. Random effect meta-analysis (Q 5030.957, df 15, I2 99.702%, p<0.001) IC50 0.645 mg / ml (95% CI: 0.557, 0.733), Standard error (0.045), p <0.001. Legend: Q = Q statistic (measure of weighted squared deviations); df =degrees of freedom; I2 (I^2) = Higgins heterogeneity index, I² <50%, heterogeneity irrelevant; I²> 75%, significant heterogeneity; C.I. confidence intervals P = P value. The graph for each study shows the cell lines investigated, the first author and the date of publication, also the measurement of the IC50 expressed in mg\ml with the confidence intervals is reported. The final value is expressed in bold with the relative confidence intervals. The red line shows the position of the average value and the diamond in light blue the measure of the average effect.
Figure 3. Forest plot of the 4 Subgroups. Subgroup Cervix: IC50 0.917 mg/mL (Hela, SiHa, Hep-2); Subgroup Lung: IC50 0.460 mg/mL (NCI-H292, A549, MRC-5); Subgroup mammary gland/breast: IC50 0.699 mg/mL (MDA-MB-231, F3II, MDA-MB-468); Subgroup colorectal, adenocarcinoma: IC50 mg/mL 0.890 (HT-29). The results of the meta-analysis for each subgroup are highlighted in bold. the yellow diamonds in the forest plot indicate average effect for each subgroup investigated. The N.A. stands for not applicable.
Round 2
Reviewer 2 Report
There is no comment about this manuscript.